# Exploration of collective tactical variables in elite netball: An analysis of team and sub-group positioning behaviours

**Ryan W. Hodder**[1]*, **Will G. Hopkins**[1], **Kevin A. Ball**[1], **Jamie Bahnisch**[2], **Fabio R. Serpiello**[1]

1 Institute for Health and Sport (IHES), Victoria University, Melbourne, Victoria, Australia, 2 Victorian Institute of Sport, Melbourne, Australia

* ryan.hodder@live.vu.edu.au

## Abstract

Collective tactical behaviours are aspects of player interactions that are particularly important in netball, due to its unique restrictions on player movement (players unable to move when in possession of the ball and positional spatial restrictions). The aim of this study was to explore variables representing collective tactical behaviours in netball. A local positioning system provided player positions of one team throughout seven elite-level netball matches. The positions were analysed to provide mean, variability (standard deviation) and irregularity (normalised approximate entropy) for each attack and defence possession (470 and 423, respectively) for the team and positional subgroups (forwards, midcourts and defenders) for 10 position-related variables. Correlational analyses showed collective tactical variables could be grouped as lateral and longitudinal dispersion variables. The variables were each analysed after log transformation with a linear mixed model to compare attack and defence and to estimate standardised effects on attack and defence of possession outcome, possession duration, score difference, match time, opposition strength and season time. During attack, the team and all sub-groups adopted greater lateral dispersion between players, while on defence there was generally greater longitudinal dispersion. The team also showed increased longitudinal dispersion when home and opposition possessions ended in a score. Additionally, greater irregularity was observed in active sub-groups (forwards on attack, defenders on defence). Score difference and opposition strength had trivial-small but generally unclear effects. In conclusion, these effects show that analysis of player positions on attack and defence is a promising avenue for coaches and analysts to modify collective tactical behaviours in netball.

## Introduction

The analysis of collective team behaviours has become more prominent over the past decade [1–5]. The focus of collective behaviour research on soccer has been afforded by the early ability to track a player's location using vision-based tracking systems and global positioning

**Funding:** RH is supported by a scholarship partially funded by Netball Victoria, which provides financial input for the Local Positioning Units used in the study. The funders had no role in study design, data collection and analysis, decision to publish, or preparation of the manuscript.

system devices in outdoor environments [6, 7]. Conversely, indoor-court sports have been limited due to inability of global positioning system devices to penetrate indoor stadium roofs and ceilings and due to the occlusion errors, which are common with vision based tracking systems [8]. However, recent advancements in Local Positioning Systems (LPS) now provide an avenue to collect player tracking data indoors [9, 10].

Tactical behaviour in sport can be defined as how individual players and teams as a whole, utilise shared affordances and environmental constraints to achieve a shared goal (e.g., scoring a goal) [4, 5, 11–13]. Affordances can arise from a player's perceptions of the environment and can dictate the available actions players can use to interact with the dynamically shifting tactical opportunities [13]. An ecological dynamics approach provides a theoretical framework, suggesting the importance of measuring tactical behaviour, to properly capture the complex and dynamic nature of a team's performance, which encompass' individual player actions and affordances that allow adaption of collective behaviours with teammates [11, 14]. To capture such behaviour, the introduction of variables providing an aggregated view of the collective and tactical behaviours of teams and sub-groups have arisen [4, 15, 16].

Inter-player distance, due to its simplicity and ease of computation, is a commonly assessed collective tactical variable, used to measure the distance between two players [17]. Literature suggests that, on average, players of the attacking team are further apart than those of the defending team [18]. The team centroid, another commonly used collective tactical variable [5, 12], is an aggregate measure (mean of x-coordinate and y-coordinate) used to display the central position of a team or group of players [2, 19, 20]. The centroid can also be used for smaller groups of players within the team (defenders, midcourts and forwards), with research suggesting that players are more coordinated with their position-specific centroid as they are adjusted to position specific affordances and constraints which helps with sub-group behaviours [21]. Additionally the width, length and length per width ratio explain how spread laterally and longitudinally a group of players are [19, 22]. Research has previously found that the team's dispersion was longer and wider during offensive phases of a match compared to defensive phases in soccer against weaker teams in Australian rules football [23, 24]. The length per width ratio has been used to explain the playing shape of a team or sub-group [25], with greater values of length per width ratio representing an elongated playing shape [25, 26]. Finally, the surface area, which explains the playing shape and space of a group of players and the stretch index, measuring the dispersion of a group of players, have also been used to analyse collective tactical behaviours [11, 27, 28].

Team-level behaviours provide a global insight into tactics, positioning and strategies, it has however been suggested that collective behaviours should also be analysed at sub-group levels. Allowing capture of different dynamics associated with individuals of sub-groups sharing similar tendencies and goals, which are separated from the global team structure [5, 29–32]. Additionally sub-groups behaviours and the individual interactions during self-organisation can cause order to develop in a complex system at a global level [5, 33]. Team-level aggregate measures provide useful information at a global level, however they may fail to capture smaller groups movements and behaviours, as well as including certain players or positions that are not always active in the play [4, 12, 29].

Furthermore, additional measures of collective tactical variables including, mean, variability (standard deviation) and irregularity (approximate entropy) are also of importance when using collective team variables. As they provide further context to the range, central point and predictability of collective tactical variables. Most research has represented collective tactical variables using the mean, providing values on the same scale as the collected variables, allowing for ease of interpretation [4, 27, 34]. A number of studies have also explored the use of the variability and irregularity when measuring collective tactical variables; both these type of

measurements provide further insight into the evolution and regularity of these variables during a possession or match [3, 35, 36]. Behaviours where irregularity is high (high approximate entropy) represent high complexity, while behaviours where regularity is high (low approximate entropy) represent low complexity and thus are more predictable [35, 37].

One sport which has yet to explore collective tactical variables is netball. Netball research has previously investigated match and training physical profiles [38–42], technical aspects utilising notational methods [43, 44] and more recently non-linear analysis and machine learning techniques to account for the dynamic nature of the sport [45–51]. However, one area of research which is yet to be studied is collective team behaviours, with a recent systematic review highlighting no such research articles having been published on netball, [52] with a number of additional studies suggesting future research is required in team and player positioning [49, 53, 54]. This is pertinent to netball, as the positional court restrictions and inability of the player in possession of the ball to move, requires a collective team performance, with previous literature stating key aspects of netball performance being off-ball space creation, maintaining unit structure, timing and support [48, 53].

Additionally, previous research in sports outside of netball has accounted for numerous contextual factors, such as possession outcome, score status, strength of opposition, possession duration, match time and season time, which may be influencing collective behaviours [23, 55–59]. When teams are losing in the score, research has shown they display lower intra-team synchronisation [57]. Decreased values of dispersion and average field position during the second half of a soccer match highlight the effect of match time [60]. Additionally, longer possessions have been associated with more regular patterns as players have more time to perceive information [37]. Finally, inter-player distance between a dyad, has been shown to be predictive of try outcomes in rugby union [14], emphasising the importance of accounting for such contextual factors when measuring collective tactical variables. These variables have yet to be studied in netball and thus provide several novel streams of research.

As such, the aim of this research study is to explore collective tactical variables in netball, at the team and sub-group levels, accounting for contextual effects, to provide a broad initial insight into the potential uses and applicability of chosen variables to measure collective behaviours in elite netball (and sub-groups).

## Methods

### Experimental overview and participants

Spatiotemporal data was collected from seven competitive elite-level netball matches, from a professional team participating in the Suncorp Super Netball League. All 12 female participants (26.9 ± 3.7 years old, 179.8 ± 4.5 cm, 67.5 ± 8.4 kg) received verbal and written information regarding the procedures of the study and provided written consent to participate in the study. The study was approved by the researchers' institutional ethics committee. All seven matches were played at an arena fitted with the ClearSky T6 Local Positioning System (Catapult, Australia) and calibrated to avoid metal interference and to ensure optimal placement for coverage of netball court dimensions. All matches were played on parqueted surface with dimensions 30.5 x 15.25 m in accordance with the International Netball Federation Rules of Netball 2016 [61]. The matches consisted of 4 x 15-min quarters with 5-min breaks between quarters and a 15-min break between halves.

### Data collection

**Positional data.** Positional data for all participants were collected using the Catapult ClearSky T6 Local Positioning System. This system has previously been validated against the

Vicon motion capture system (©Vicon Motion Systems, Oxford Metrics, UK), with a mean positional estimate difference of 0.21 ± 0.13 m in an optimal setup [62] and mean bias between 0.2–12% for measurement of total distance, mean and peak speed, and mean and peak accelerations during linear drills [63]. Additionally for tactical use, inter-unit distance measurement was found to have a root mean square error of 0.20 ± 0.05 m [64]. The same calibration and setup was used for the LPS as in the validation study by [64], as it is a fixed system in the stadium. Each participants match uniforms were fitted with an internal stitched pouch, which housed the LPS receiver tag positioned between the shoulder blades. Positional data was captured at 10 Hz and downloaded in the software-derived format as .csv files to the LPS data processing laptop immediately after each match.

**Event data.** Match event data were collected post-event using video analysis software Sportscode version 11.3.0 (Hudl Sportscode, Nebraska, United States of America), by the first author. The first author recorded event data post-match for all seven matches, including team in possession and score difference (numeric difference between attacking team score and opposition team score). Possession was defined as the team who had possession of the ball, either attacking team (team participating in study) or opposition team. Possessions began with the umpire's signal, indicating the commencement of a centre pass or restart after a penalty. Additionally, possessions also began in open play through a turnover (either general play turnover or unforced turnover) [65], in which possession started as the closest $10^{th}$ of a second to which an attacking or opposition team player gained possession of the ball. During timeouts, pauses in play or when neither team had clear possession of the ball, the time phase was coded as "No Possession" [66, 67]. Possession outcome was recorded as either a score or turnover for the home team (score count = 311 and turnover count = 160) or opposition team (score count = 297 and turnover count = 166), with any possession not ending in a score or turnover recorded as "other" (n = 10). Other could include instances where end of a quarter ends the possession, or a stoppage in play for injury that is neither a turnover or a score. Score difference was also recorded for each possession, as a numeric value indicating the difference between the attacking team score and opposition score. Finally, two additional variables were added to the data set; possession duration and ladder-points difference (difference between team and oppositions end of season ladder points), to account for possession duration and opposition strength differences respectively. Ladder-points at the end of a season has previously been used as a measure of opposition strength [23, 58], in the current study ladder-points is modelled as a difference between the ladder points of the team in the study and their opposition (The top ranked team had a ladder point of 83, the bottom ranked 26 and the analysed team a ladder point of 63). The first author's inter-rater agreements score was 95.2%.

## Data processing

Positional data were processed first in R statistical software (R, Vienna, Austria), version 4.0.2. Utilizing the follow packages; tidyverse, gganimate, transformr, pracma, plyr, purrr, reshape and geometry. The origin for player position was set to one corner of the netball court. Attacking direction was reset each quarter so the analysed team's coordinates faced to the left always (as teams change the direction after each break). Upon completion of initial processing, player position data and event data were synchronised using a two-step procedure. First, event data were converted to milliseconds and aligned with positional data. Secondly, datasets were synced using a common synchronisation point; when the centre position player first made contact with the centre circle at the start of each quarter and restart after time-outs. This step was adjusted to a tenth of a second precision based on the plotting animations of player positional data [36]. Additionally, when a player was substituted, the new player's positional data

were added to the previous player's data at the point of substitution (n = 35) [22]. Furthermore, the study methodology was written following a recently published protocol [68] in order to warrant the strict description of the use of technology, scoring 16 out of 21 points (76%).

A value for each of ten collective tactical variables was calculated every 0.1 s for the whole team and for each sub-group: forwards (goal shooter, goal attack, wing attack), midcourts (wing attack, centre, wing defence), and defenders (wing defence, goal defence, goal keeper). These positional sub-groups were selected to mirror previous research [69], the collective tactical variables were defined as follows (see Figs 1–4 for an additional illustration of the first five collective tactical variables for the team and sub-groups):

**Mean inter-player distance.** Mean distance of each player to all other players in the team or sub-group, including each player-player dyad only once in the calculation (21 unique dyads) [70].

**Centroid.** Average cartesian coordinates of the team or sub-group of players [3, 19]. The centroid's longitudinal and lateral components were also calculated to provide further information regarding average positioning on the court.

**Length.** Distance between the most backward and forward players at each end of the court [19].

**Width.** Distance between the two most lateral players on each side of the court [19, 67].

**Width per length ratio.** Dividing width by length, with values >1 indicating a wider playing shape and <1 a more elongated shape.

**Surface area.** Area ($m^2$) of the smallest convex hull of boundary players [19, 71, 72].

**Stretch index.** Distance of each player to the teams or sub-groups centroid [11, 28, 72]. Similar to the centroid, the stretch index was also separated into longitudinal and lateral values, representing the amount of dispersion in either direction.

The mean, standard deviation (SD) and normalised approximate entropy of each collective tactical variable was calculated for each possession. Approximate entropy is a measure used on time-series data to assess the randomness and regularity of complex systems [73]. Approximate Entropy (m, r, N) can be used to assess the probability that a pattern of segmented data in a time series (N) is able to predict the pattern of segmented data of the same time series a certain distance apart. Two fixed parameters m (length of segment to be compared) and r (tolerance factor) were set at 2 and 0.2 respectively, following suggestions from Stergiou et al. [74] in which further explanation of input calculations can be found. Due to each possession in this data set varying in length, normalised values of ApEn where computed, to allow for comparison of ApEn values between all possessions of different lengths. Normalised ApEn was calculated using the ApEn ratio-random method, whereby the original ApEn value is divided by the mean ApEn value of 100 randomly scrambled values of the time series [75]. In this ratio, values close to 0 represent high regularity while values close to 1.0 represent low regularity [75]. A total of 898 attack and defence possessions were used in the analysis (see S1 and S2 Videos for moving illustration of collective tactical variables for the team and sub-groups), with an average duration of 18.2 ± 8.1 s (mean ± SD). From this point forward, the mean, standard deviation and approximate entropy will be referred to as "measures" when referenced collectively.

## Statistical analysis

Analyses were conducted with the Statistical Analysis System (Studio On Demand for Academics, version 9.4, SAS Institute, Cary NC). The mean, standard deviation and ApEn for each collective tactical variable were log-transformed for further analysis (with the exception of the mean for centroid longitudinal and lateral, for which log-transformation would not be appropriate). Pearson correlations between all the collective tactical variables were first derived

# Team

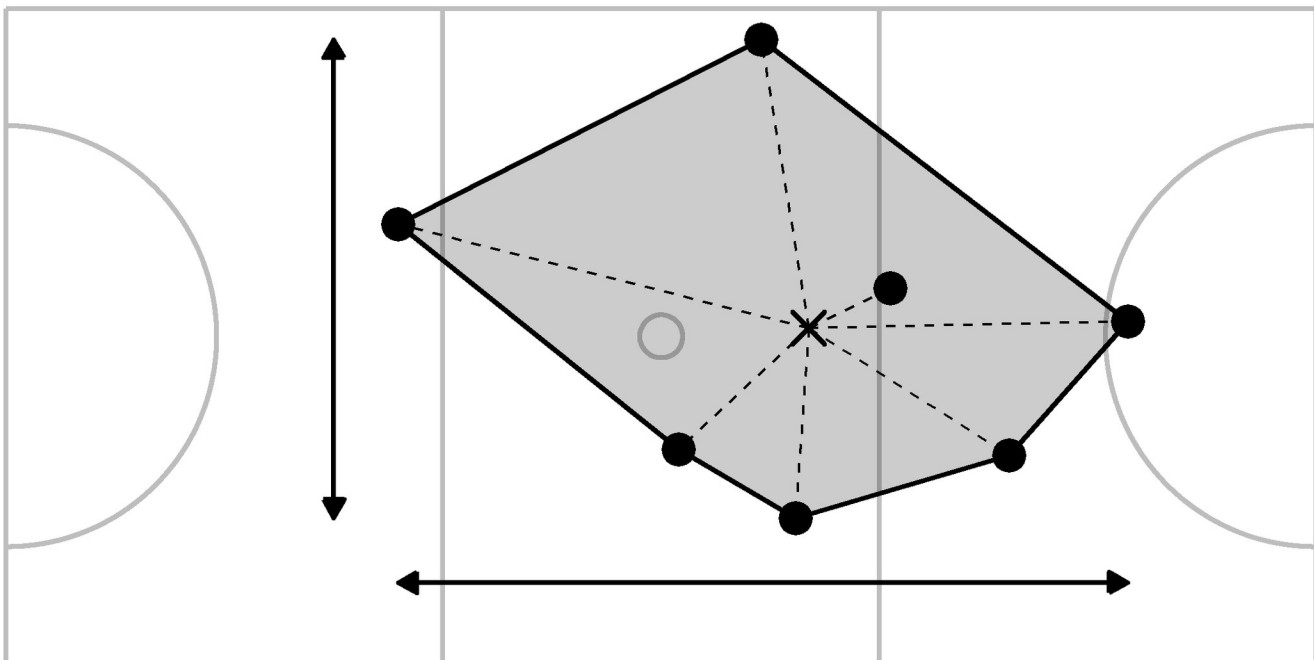

**Fig 1. Illustration of five collective tactical variables for the team.** Players positions (•) are combined to provide the centroid (mean position, ×), length (horizontal arrow line), width (vertical arrow line), stretch index (mean length of dashed lines) and surface area (grey shading). Solid grey lines represent court markings and define, for example, the left- and right-hand thirds of the court area to which the goal keeper and goal shooter are restricted.

to assess the similarity of the variables. For presentation purposes, the set of three correlations (for the mean, standard deviation and approximate entropy) are shown for each pair of collective tactical variables in a correlation matrix for the team (Tables 1 and 2) and each positional subgroup [S1–S6 Tables]. The order of the variables in each correlation matrix was changed to reveal clusters of similar variables (higher correlations within clusters than between clusters). Regardless of the correlations, each variable was subsequently analysed as the dependent in a general linear mixed model (Proc Mixed in SAS). There was a total of four zero or negative ApEn values; these were set to half the minimum positive value, before log transformation. The aim of each analysis was to estimate factors modifying the collective tactical variable separately for attack and for defence, but to include the factors interacted with possession type so that the differences between attack and defence and the uncertainty of the differences could be estimated.

The fixed effects in the model were possession type (to estimate mean values of the collective tactical variable in attack and defence) and possession type interacting with each of the following variables (to estimate their modifying effect in attack and defence): possession outcome (home team score or turnover on attack and opposition team score or turnover on defence), possession duration (linear numeric, log-transformed), score difference (linear numeric), match time (linear numeric), ladder-points difference (linear numeric), and finally match number (linear numeric). Visual inspection of plots of residuals vs predicteds and residuals vs predictors showed no obvious evidence of non-uniformity and non-linearity of effects. The magnitudes of the effects of the nominal predictors (possession type, possession outcome) were evaluated for the mean possession duration (17 s for attack, 16 s for defence), zero score

## Midcourts

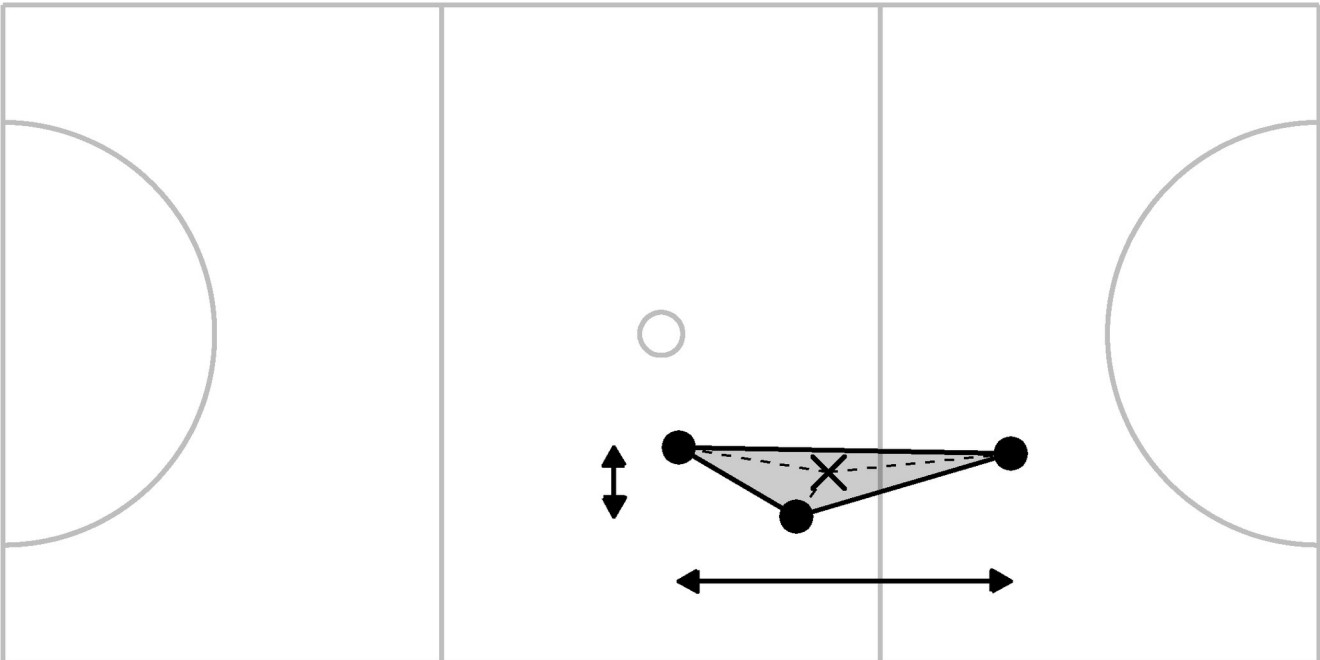

**Fig 2. Illustration of five collective tactical variables for the midcourt sub-group.** Players positions (•) are combined to provide the centroid (mean position, ×), length (horizontal arrow line), width (vertical arrow line), stretch index (mean length of dashed lines) and surface area (grey shading). Solid grey lines represent court markings and define, for example, the left- and right-hand thirds of the court area to which the goal keeper and goal shooter are restricted.

difference (representing equally matched teams during the match), the middle of a 60-min match, zero ladder-points difference (representing equally matched teams across the season), and the middle match of the season. The magnitudes of the linear numeric modifying effects were evaluated as follows: for possession duration the effect of two SDs (factor increases of 2.4 on attack and 2.7 on defence), for score difference 10 points (representing approximately two SDs of the score-difference distribution), for match time 60 min (representing the match trend), for ladder-points difference 55 (for the effect of the strongest opposition minus the weakest opposition), and for match number 13 (representing the season trend between the first and fourteenth match). Only the seven home matches were analysed.

The random effects were match identity (to adjust for changes in means of the collective tactical variable between matches) and the residual (representing changes between possessions not explained by all the other effects in the model). Separate variances were specified for the random effects on attack and on defence; for the residual, separate variances were also specified for each quintile of possession duration within possession type (to allow for the possibility of substantial differences in error with different possession durations, given the strong fixed effects of possession duration that were observed for ApEn). Outliers identified by standardised residuals >4.5 (up to five of the 898 possessions) were deleted, then the data were reanalysed [76].

Means and SDs of the collective tactical variables were derived from the mixed model: the means are back-transformed least-squares means; the SDs are the residuals for the middle quintile for possession duration, back-transformed to percent units. Effects are presented in

# Forwards

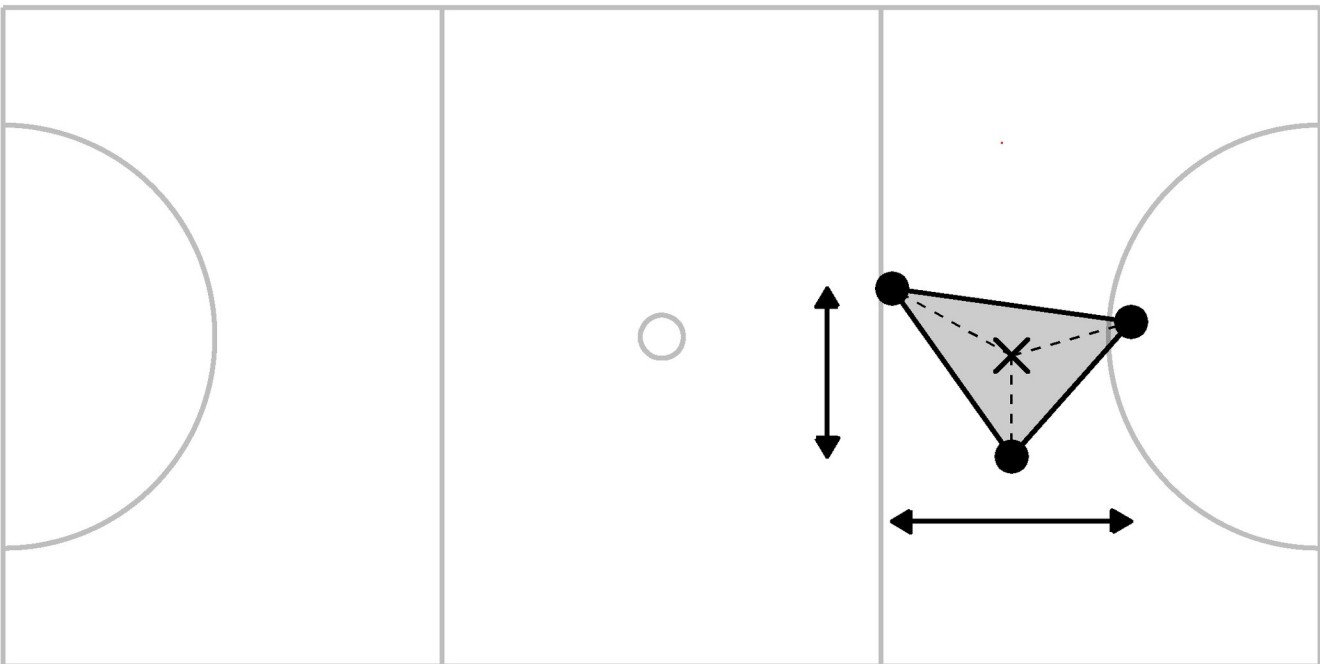

**Fig 3. Illustration of five collective tactical variables for the forward sub-group.** Players positions (•) are combined to provide the centroid (mean position, ×), length (horizontal arrow line), width (vertical arrow line), stretch index (mean length of dashed lines) and surface area (grey shading). Solid grey lines represent court markings and define, for example, the left- and right-hand thirds of the court area to which the goal keeper and goal shooter are restricted.

percent units with uncertainty expressed as ±90% compatibility limits. Magnitudes of effects were assessed using standardisation, where the standardising SD was the square root of the mean of all the residual variances. For those who prefer a frequentist interpretation of sampling uncertainty, decisions about magnitudes accounting for the uncertainty were based on one-sided interval hypothesis tests, according to which a hypothesis of a given magnitude (substantial, non-substantial) is rejected if the 90% compatibility interval falls outside that magnitude [77], p values for the tests were therefore the areas of the sampling t distribution of the effect falling in the hypothesized magnitude, with the distribution centered on the observed effect. Hypotheses of inferiority (substantial negative) and superiority (substantial positive) were rejected if their respective p values ($p_-$ and $p_+$) were <0.05; rejection of both hypotheses represents a decisively trivial effect in equivalence testing. The hypothesis of non-inferiority (non-substantial-negative) or non-superiority (non-substantial-positive) was rejected if its p value ($p_{N-} = 1 - p_-$ or $p_{N+} = 1 - p_+$) was <0.05, representing a decisively substantial effect in minimal-effects testing. A complementary Bayesian interpretation of sampling uncertainty is also provided, when at least one substantial hypotheses was rejected: the p value for the other hypothesis is the posterior probability of a substantial true magnitude of the effect in a reference-Bayesian analysis with a minimally informative prior [78], and it was interpreted qualitatively using the following scale: >0.25, possibly; >0.75, likely; >0.95, very likely; >0.995, most likely [76]. The probability of a trivial true magnitude ($1 - p_- - p_+$) was also interpreted with the same scale. Probabilities were not interpreted for unclear effects: those with inadequate precision at the 90% level, defined by failure to reject both substantial hypotheses ($p_- > 0.05$ and $p_+ > 0.05$). Effects on magnitudes and probabilities of a weakly informative normally

# Defenders

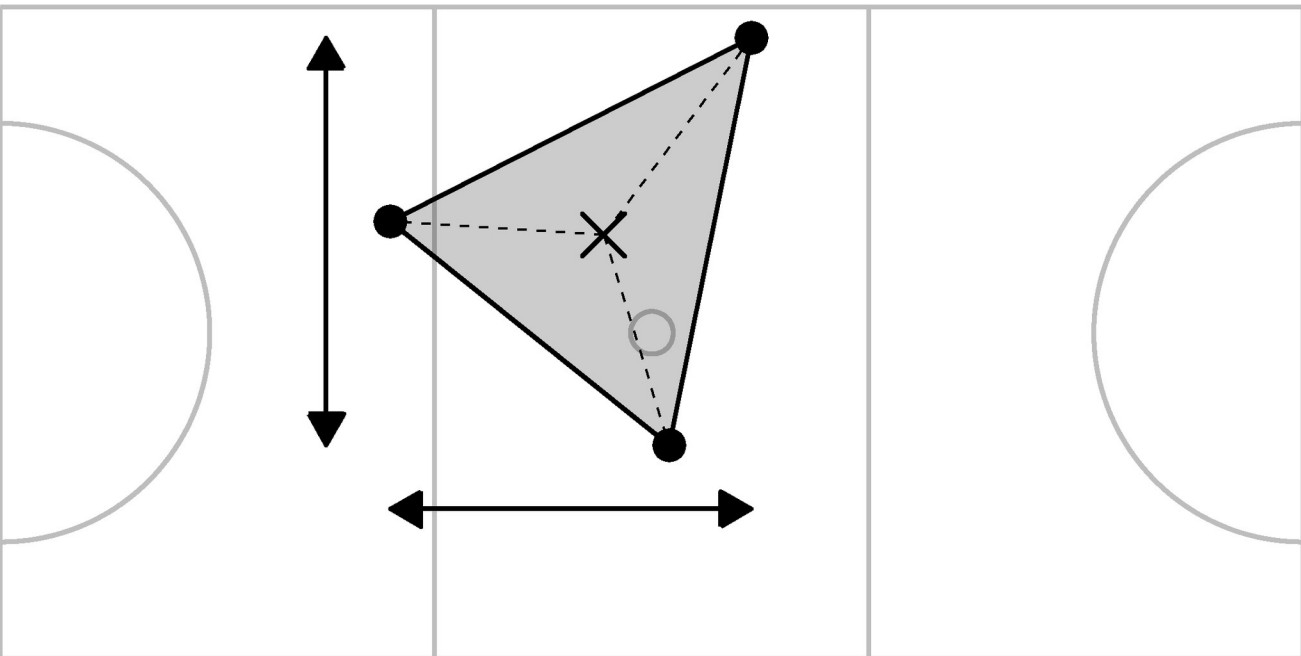

**Fig 4. Illustration of five collective tactical variables for the defender sub-group.** Players positions (•) are combined to provide the centroid (mean position, ×), length (horizontal arrow line), width (vertical arrow line), stretch index (mean length of dashed lines) and surface area (grey shading). Solid grey lines represent court markings and define, for example, the left- and right-hand thirds of the court area to which the goal keeper and goal shooter are restricted.

distributed prior centered on the zero effect and excluding extremely large effects at the 90% level were also investigated [78, 79]. Effects with adequate precision at the 99% level ($p_- < 0.005$ or $p_+ < 0.005$) are highlighted in bold in tables, since these represent stronger evidence against substantial hypotheses than the 90% level and therefore incur lower inflation of error with multiple hypothesis tests.

## Results

### Correlations between collective tactical variables

Tables 1 and 2 show the pairwise correlations between all the variables, ordered to reveal two clusters of variables with generally higher correlations within the clusters than between the clusters. The clustering is the same and the correlations have similar values for attack and defence. The first cluster can be interpreted as dispersion of the players in the longitudinal direction, the second as dispersion in the lateral direction, and the two centroids as the mean position in the longitudinal and lateral directions. Variables showing pairwise correlations of greater than 0.82 in the first cluster are stretch index, inter-player distance and stretch index longitudinal, while those in the second cluster are width and stretch index lateral.

For positional sub-groups [see S1–S6 Tables], the clusters were less well defined, owing to somewhat higher correlations between some variables between clusters. When considering correlations of at least ∼0.8 within clusters, the clusters were the same as those for the team, although two of the three correlations were <0.5 for defenders on defence.

**Table 1. Correlations between all collective tactical variables for the team on attack.** Each set of three values are correlations for the mean, variability and irregularity respectively. The variables have been ordered and outlined to show clusters with generally higher correlations between variables within the clusters than between the clusters.

| Variable | 1 | 2 | 3 | 4 | 5 | 6 | 7 | 8 | 9 | 10 |
|---|---|---|---|---|---|---|---|---|---|---|
| 1. Stretch index | | .97, .95, .93 | .94, .90, .89 | .72, .52, .60 | .06, .18, .48 | .05, .14, .47 | -.24, .33, .43 | .43, .19, .49 | -.01, .23, .24 | .01, .13, .36 |
| 2. Inter-player distance | .97, .95, .93 | | .89, .82, .82 | .81, .61, .67 | .11, .18, .46 | .10, .13, .44 | -.23, .31, .42 | .51, .23, .50 | -.01, .25, .28 | .00, .14, .33 |
| 3. Stretch index longitudinal | .94, .90, .89 | .89, .82, .82 | | .73, .55, .62 | -.24, .26, .50 | -.26, .23, .50 | -.47, .42, .49 | .14, .14, .42 | -.02, .20, .26 | .00, .12, .42 |
| 4. Length | .72, .52, .60 | .81, .61, .67 | .73, .55, .62 | | -.19, .13, .47 | -.17, .11, .46 | -.53, .36, .52 | .27, .07, .45 | .05, .30, .27 | .00, .20, .39 |
| 5. Width | .06, .18, .48 | .11, .18, .46 | -.24, .26, .50 | -.19, .13, .47 | | .89, .75, .82 | .88, .62, .78 | .83, .69, .77 | .00, .17, .25 | .03, .19, .48 |
| 6. Stretch index lateral | .05, .14, .47 | .10, .13, .44 | -.26, .23, .50 | -.17, .11, .46 | .89, .75, .82 | | .79, .50, .73 | .80, .62, .76 | .04, .21, .27 | -.07, .21, .44 |
| 7. Width per length ratio | -.24, .33, .43 | -.23, .31, .42 | -.47, .42, .49 | -.53, .36, .52 | .88, .62, .78 | .79, .50, .73 | | .58, .32, .63 | -.02, .14, .24 | .02, .03, .40 |
| 8. Surface area | .43, .19, .49 | .51, .23, .50 | .14, .14, .42 | .27, .07, .45 | .83, .69, .77 | .80, .62, .76 | .58, .32, .63 | | .03, .15, .28 | .01, .19, .40 |
| 9. Centroid longitudinal | -.01, .23, .24 | -.01, .25, .28 | -.02, .20, .26 | .05, .30, .27 | .00, .17, .25 | .04, .21, .27 | -.02, .14, .24 | .03, .15, .28 | | -.02, .25, .22 |
| 10. Centroid lateral | .01, .13, .36 | .00, .14, .33 | .00, .12, .42 | .00, .20, .39 | .03, .19, .48 | -.07, .21, .44 | .02, .03, .40 | .01, .19, .40 | -.02, .25, .22 | |

Uncertainty (90% compatibility limits): ∼±0.1 to ∼±0.01 for correlations of 0.00 to 0.95 respectively assuming a sample size of ∼300.

**Table 2. Correlations between all collective tactical variables for the team on defence.** Each set of three values are correlations for the mean, variability and irregularity respectively. The variables have been ordered and outlined to show clusters with generally higher correlations between variables within the clusters than between the clusters.

| Variable | 1 | 2 | 3 | 4 | 5 | 6 | 7 | 8 | 9 | 10 |
|---|---|---|---|---|---|---|---|---|---|---|
| 1. Stretch index | | .99, .95, .92 | .98, .96, .93 | .79, .65, .74 | .03, .12, .32 | -.02, .13, .32 | -.30, .27, .41 | .45, .24, .40 | .08, .15, .14 | .10, .12, .35 |
| 2. Inter-player distance | .99, .95, .92 | | .95, .89, .86 | .85, .75, .80 | .08, .16, .35 | .04, .18, .33 | -.28, .26, .44 | .52, .32, .44 | .07, .20, .18 | .09, .15, .33 |
| 3. Stretch index longitudinal | .98, .96, .93 | .95, .89, .86 | | .79, .64, .71 | -.16, .13, .30 | -.20, .14, .32 | -.46, .31, .42 | .28, .18, .36 | .02, .1 5, .17 | .08, .15, .37 |
| 4. Length | .79, .65, .74 | .85, .75, .80 | .79, .64, .71 | | -.13, .17, .33 | -.12, .19, .29 | -.49, .27, .43 | .36, .29, .43 | -.09, .21, .20 | .05, .20, .32 |
| 5. Width | .03, .12, .32 | .08, .16, .35 | -.16, .13, .30 | -.13, .17, .33 | | .91, .77, .76 | .88, .57, .74 | .82, .69, .73 | .16, .24, .25 | .06, .26, .42 |
| 6. Stretch index lateral | -.02, .13, .32 | .04, .18, .33 | -.20, .14, .32 | -.12, .19, .29 | .91, .77, .76 | | .83, .46, .66 | .78, .66, .68 | .11, .32, .34 | .04, .31, .40 |
| 7. Width per width ratio | -.30, .27, .41 | -.28, .26, .44 | -.46, .31, .42 | -.49, .27, .43 | .88, .57, .74 | .83, .46, .66 | | .55, .28, .56 | .14, .09, .32 | .01, .04, .41 |
| 8. Surface area | .45, .24, .40 | .52, .32, .44 | .28, .18, .36 | .36, .29, .43 | .82, .69, .73 | .78, .66, .68 | .55, .28, .56 | | .07, .25, .29 | .12, .20, .38 |
| 9. Centroid longitudinal | .08, .15, .14 | .07, .20, .18 | .02, .15, .17 | -.09, .21, .20 | .16, .24, .25 | .11, .32, .34 | .14, .09, .32 | .07, .25, .29 | | -.05, .35, .32 |
| 10. Centroid lateral | .10, .12, .35 | .09, .15, .33 | .08, .15, .37 | .05, .20, .32 | .06, .26, .42 | .04, .31, .40 | .01, .04, .41 | .12, .20, .38 | -.05, .35, .32 | |

Uncertainty (90% compatibility limits): ∼±0.1 to ∼±0.01 for correlations of 0.00 to 0.95 respectively assuming a sample size of ∼300.

## Effect of attack vs defence

The means and variability of the collective tactical variables for attack and defence, and the differences between the means, are shown in Table 3. For variables within the same cluster, the means in raw units for attack (and defence) sometimes differ widely between variables (exception: means for irregularity, which are all in the same dimensionless units), but the variability in percent units, the effects in percent units, and the qualitative magnitudes of the effects are generally similar (e.g., width and stretch index lateral). With the exception of the extremely large effect on the mean for centroid longitudinal (representing the team being closer to the scoring goal on attack), the biggest effects occur in the second cluster, especially for mean and variability of width and stretch index lateral (representing a more expanded dispersion of the players on attack).

For the positional sub-groups [see S7–S34 Tables], patterns in the effects were similar to those for the team, in that highly correlated variables within clusters had similar magnitudes for the differences between attack and defence. Similar to the team, the midcourt sub-group showed its largest effects in its second cluster of variables. In contrast, the forward and defender sub-groups showed the largest differences between attack and defence in their first cluster, with the forwards having a lower value on attack and the defenders having a lower value on defence. Both forward and defender sub-groups also displayed noticeable differences in irregularity for most variables, presenting opposite sign but similar magnitude to that of the mean.

## Effect of possession outcome

Table 4 shows the difference between possessions resulting in a score vs a turnover for each of the derived measures of each collective tactical variable for the team. The first cluster of variables (representing longitudinal dispersion) for the mean and variability displayed the largest effects during attack and defence, but they were at most small. Specifically, when the home team scored, the stretch index and stretch index longitudinal showed small positive effects compared with when the possessions ended in a turnover. For possessions when the opposition scored, all the variables in the first cluster showed small positive effects. Effects on irregularity were similar on attack and defence: small and possibly or likely substantial reductions in irregularity for variables in the first cluster, but reasonable evidence of trivial effects otherwise.

Positional sub-groups displayed varied effects of possession outcome [S12 to S14 Tables]. The defender sub-group displayed the most dissimilar effects to that of the team, with its largest effects in the second cluster for mean, variability and irregularity on both attack and defence. The forward sub-group showed largest effects for the first-cluster variables, however these were opposite in sign to the team. The only positional sub-group showing effects on scoring vs turnover similar to those of the whole team was the midcourt, for irregularity of variables in the first cluster on attack.

## Effect of possession duration

The team [Figs 5–7 and S15 Table] and sub-groups [S16–S18 Tables] showed similar effects for variability and irregularity of all variables in the magnitude of the linear effect of two standard deviations of possession duration. There were large decreases for irregularity during attack and defence, while for variability there were moderate increases. For the mean of the first cluster variables, the team and midcourt displayed mostly small positive effects over the possession. For the defender and forwards sub-groups, small to moderate effects were also displayed during defence, be that a reduction for defenders and an increase for forwards.

**Table 3. Collective tactical variables for the team's attack and defence possessions.** Adjusted for possession duration, score difference, ladder point's difference, match trend and season trend. Solid lines separate the variables into the clusters defined by the correlations in Table 2. With the exception of the mean centroid longitudinal and lateral, the statistics were derived via log-transformation, hence SDs are shown as percents. Data for attack and defence are predicted means (raw units) from the mixed model, and SDs are an appropriate residual representing differences between possessions. Data for attack minus defence are predicted mean differences (% units) with 90% compatibility limits (% units) and decisions about the magnitude of the differences.

| Variables | Attack | Defence | Attack minus Defence | Magnitude |
|---|---|---|---|---|
| **Mean** | | | | |
| Stretch Index (m) | 6.78 ± 8% | 6.64 ± 10% | 2.0, ±3.8% | small↑*[0] |
| Inter-player distance (m) | 10.25 ± 7% | 9.98 ± 9% | 2.7, ±3.0% | small↑** |
| Stretch index longitudinal (m) | 6.03 ± 10% | 6.10 ± 13% | -1.2, ±4.6% | trivial |
| Length (m) | 22.0 ± 7% | 21.7 ± 8% | 1.1, ±2.8% | trivial |
| Width (m) | 7.3 ± 16% | 6.1 ± 18% | 17.6, ±3.1% | **moderate↑****** |
| Stretch index lateral (m) | 2.02 ± 16% | 1.67 ± 16% | 20.8, ±3.6% | **moderate↑****** |
| Width per length ratio | 0.30 ± 23% | 0.26 ± 22% | 15.4, ±6.7% | **moderate↑***** |
| Surface area (m$^2$) | 86 ± 17% | 73 ± 17% | 17.4, ±3.4% | **moderate↑****** |
| Centroid longitudinal (m) | 18.3 ± 1.3 | 11.7 ± 1.7 | 6.84, ±0.69 | **extremely large↑****** |
| Centroid lateral (m) | 7.5 ± 1.1 | 7.5 ± 1.1 | 0.04, ±0.26 | trivial |
| **Variability** | | | | |
| Stretch index (m) | 0.82 ± 42% | 0.88 ± 46% | -6.8, ±9.2% | trivial↓[0]* |
| Inter-player distance (m) | 1.00 ± 44% | 1.09 ± 45% | -8.1, ±9.0% | trivial↓[0]* |
| Stretch index longitudinal (m) | 0.99 ± 50% | 0.97 ± 46% | 1, ±13% | trivial |
| Length (m) | 2.09 ± 43% | 2.19 ± 48% | -4.4, ±9.3% | trivial↓[0]* |
| Width (m) | 1.70 ± 33% | 1.34 ± 43% | 23.4, ±6.2% | **moderate↑****** |
| Stretch index lateral (m) | 0.50 ± 36% | 0.38 ± 41% | 28.1, ±6.7% | **moderate↑****** |
| Width per length ratio | 0.97 ± 69% | 1.00 ± 50% | -3, ±12% | trivial[00] |
| Surface area (m$^2$) | 19.6 ± 41% | 17.4 ± 48% | 10.8, ±6.0% | **small↑**** |
| Centroid longitudinal (m) | 1.95 ± 49% | 1.50 ± 60% | 30, ±14% | **moderate↑***** |
| Centroid lateral (m) | 0.82 ± 55% | 0.68 ± 61% | 18.2, ±9.7% | **small↑**** |
| **Irregularity** | | | | |
| Stretch index | 0.19 ± 56% | 0.16 ± 70% | 16.1, ±8.3% | **small↑**** |
| Inter-player distance | 0.20 ± 56% | 0.17 ± 62% | 14.7, ±9.0% | **small↑**** |
| Stretch index longitudinal | 0.17 ± 57% | 0.15 ± 75% | 8, ±12% | small↑[0] |
| Length | 0.20 ± 54% | 0.18 ± 72% | 7, ±14% | trivial[0] |
| Width | 0.34 ± 44% | 0.38 ± 44% | -12.0, ±4.8% | **small↓**** |
| Stretch index lateral | 0.31 ± 36% | 0.38 ± 31% | -19.7, ±4.5% | **small↓****** |
| Width per length ratio (m) | 0.33 ± 66% | 0.27 ± 46% | 32, ±12% | **moderate↑****** |
| Surface area | 0.35 ± 43% | 0.34 ± 46% | 1.9, ±6.5% | trivial[00] |
| Centroid longitudinal | 0.08 ± 86% | 0.10 ± 98% | -18, ±20% | small↓*[0] |
| Centroid lateral | 0.20 ± 47% | 0.23 ± 61% | -16, ±12% | **small↓**** |

90%CL, 90% compatibility limits; ↑, increase; ↓, decrease.

Magnitudes are based on the following scale for standardised changes in the mean: <0.2, trivial; 0.2–0.6, small; 0.6–1.2, moderate; 1.2–2.0, large; 2.0–4.0, very large; >4.0 extremely large

Reference-Bayesian likelihoods of substantial change: *possibly; **likely; ***very likely, ****most likely.

*** and **** indicate rejection of the non-superiority or non-inferiority hypothesis ($p_{N-}$ or $p_{N+}$ <0.05 and <0.005 respectively).

Reference-Bayesian likelihoods of trivial change: [0]possibly; [00]likely.

Likelihoods are not shown for effects with inadequate precision at the 90% level (failure to reject any hypotheses: p>0.05).

Effects in **bold** have adequate precision at the 99% level (p<0.005).

**Table 4. Difference between home score and home turnover (on attack) and between opposition score and opposition turnover (on defence).** For each of the derived measures of each collective tactical variable for the **team**. With the exception of the mean centroid longitudinal and lateral, the statistics were derived via log-transformation, hence data are the predicted changes (%, ±90% compatibility limits) and decisions about the magnitude of the changes.

| Variables | Home (attack) | | Opposition (defence) | |
|---|---|---|---|---|
| | Score–Turnover | Decision | Score–Turnover | Decision |
| **Mean** | | | | |
| Stretch index (m) | 2.0, ±1.3% | **small↑*0** | 2.9, ±1.8% | **small↑** |
| Inter-player distance (m) | 1.4, ±1.1% | **trivial↑0*** | 1.9, ±1.5% | **small↑*0** |
| Stretch index longitudinal (m) | 3.6, ±1.6% | **small↑** | 4.4, ±2.2% | **small↑*** |
| Length (m) | 1.6, ±1.2% | **trivial↑0*** | 2.6, ±1.4% | **small↑** |
| Width (m) | -4.8, ±2.5% | **small↓** | -5.3, ±2.6% | **small↓** |
| Stretch index lateral (m) | -4.1, ±2.6% | **small↓*0** | -4.9, ±2.7% | **small↓** |
| Width per length ratio | -6.2, ±3.3% | **small↓** | -7.5, ±3.2% | **small↓*** |
| Surface area (m²) | -2.3, ±2.6% | **trivial00** | -2.4, ±2.8% | **trivial↓0*** |
| Centroid longitudinal (m) | 0.62, ±0.24 | **small↑** | -0.59, ±0.28 | **small↓*** |
| Centroid lateral (m) | 0.09, ±0.19 | trivial00 | -0.05, ±0.17 | trivial00 |
| **Variability** | | | | |
| Stretch index (m) | 12, ±7.2% | **small↑** | 17, ±8.5% | **small↑*** |
| Inter-player distance (m) | 11, ±6.7% | **small↑*0** | 18, ±8.8% | **small↑*** |
| Stretch index longitudinal (m) | 14, ±7.3% | **small↑** | 18, ±8.3% | **small↑*** |
| Length (m) | 11, ±6.8% | **small↑** | 18, ±8.4% | **small↑*** |
| Width (m) | -3.4, ±5.1% | **trivial00** | -5.0, ±5.9% | **trivial↓0*** |
| Stretch index lateral (m) | -4.7, ±5.2% | **trivial00** | -3.2, ±5.7% | **trivial00** |
| Width per length ratio | -9.1, ±7.7% | **trivial↓0*** | -10.1, ±7.6% | **small↓*0** |
| Surface area (m²) | -3.4, ±5.3% | **trivial00** | -2.9, ±6.0% | **trivial00** |
| Centroid longitudinal (m) | 3.4, ±6.9% | **trivial00** | 13, ±8.8% | **small↑** |
| Centroid lateral (m) | -8.1, ±6.6% | **trivial↓0*** | -0.4, ±8.6% | trivial00 |
| **Irregularity** | | | | |
| Stretch index | -16, ±6.9% | **small↓** | -13, ±8.6% | **small↓*0** |
| Inter-player distance | -15, ±6.8% | **small↓** | -9, ±8.5% | **trivial↓0*** |
| Stretch index longitudinal | -17, ±6.8% | **small↓** | -12, ±8.5% | small↓*0 |
| Length | -13, ±6.8% | **small↓*0** | -11, ±8.7% | **small↓*0** |
| Width | -7, ±5.3% | **trivial↓0*** | -2.2, ±5.8% | **trivial00** |
| Stretch index lateral | -2.9, ±5.5% | **trivial00** | -1.4, ±5.4% | trivial00 |
| Width per length ratio | 13, ±9.0% | **small↑** | 6.6, ±7.6% | **trivial↑0*** |
| Surface area | -3.9, ±5.4% | **trivial00** | 0.00, ±6.5% | trivial00 |
| Centroid longitudinal | -3.8, ±10% | **trivial00** | -18, ±10% | **small↓** |
| Centroid lateral | -1.9, ±7.0% | **trivial00** | 2.9, ±8.3% | trivial00 |

↑, increase

↓, decrease.

Magnitudes are based on the following scale for standardized changes in the mean: <0.2, trivial; 0.2–0.6, small; 0.6–1.2, moderate; 1.2–2.0, large; 2.0–4.0, very large; >4.0 extremely large

Reference-Bayesian likelihoods of substantial change:

*possibly

**likely

***very likely.

*** indicates rejection of the non-superiority or non-inferiority hypothesis ($p_{N-}$ or $p_{N+}$ <0.05).

Reference-Bayesian likelihoods of trivial change: [0]possibly; [00]likely.

Likelihoods are not shown for effects with inadequate precision at the 90% level (failure to reject any hypotheses: p>0.05).

Effects in **bold** have adequate precision at the 99% level (p<0.005).

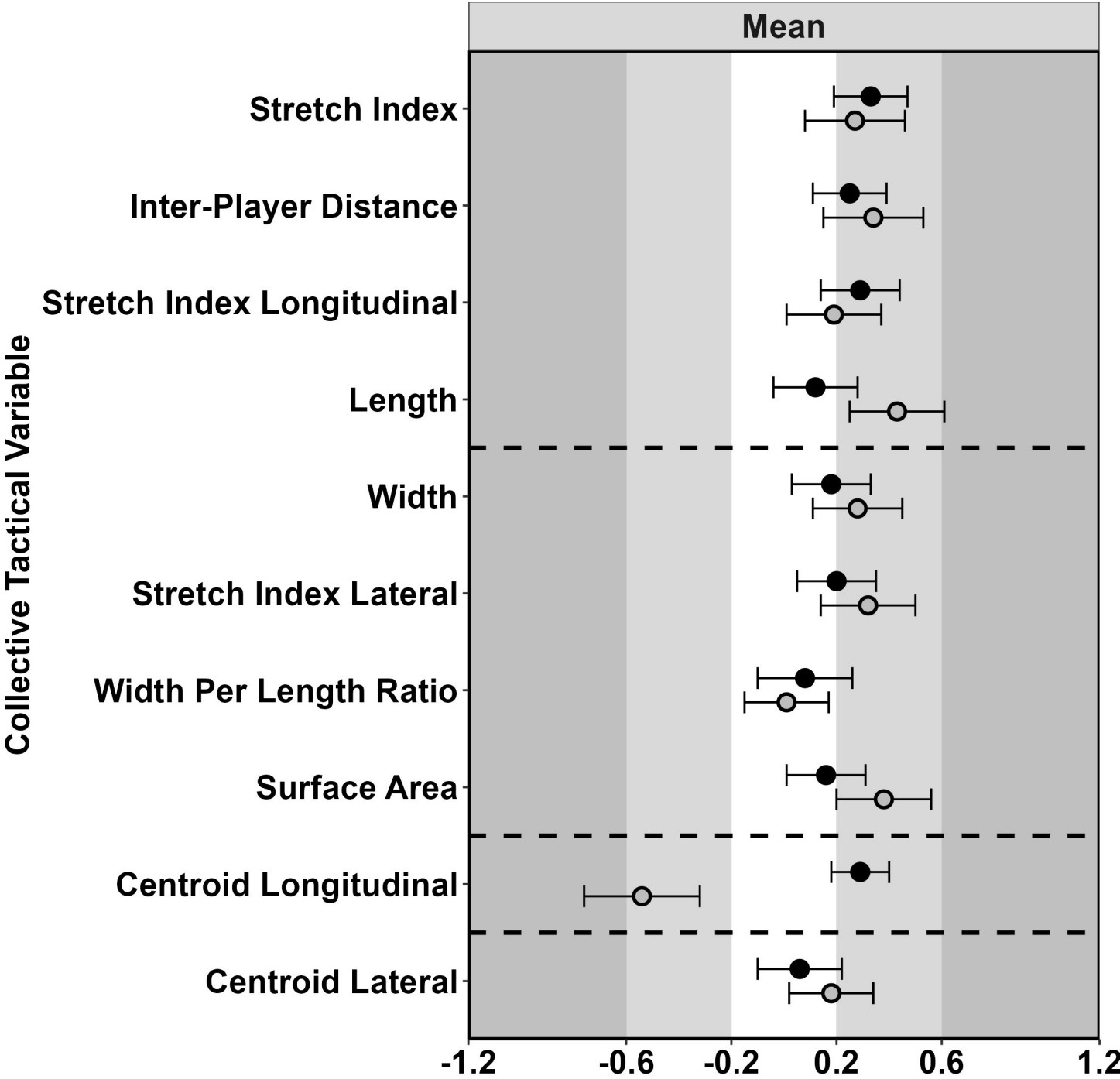

**Fig 5. Standardised effect of two SDs of possession duration for the mean on collective tactical variables.** For the team on attack (filled circles) and defence (open circles). Circles towards the left indicate higher in short possession and the right higher in long possession.

### Effect of score difference

The majority of the observed effects of 10 points of score difference were trivial [S19–S22 Tables], but only a few variables were decisively trivial and none was decisively substantial. In contrast, for all positional sub-groups, the mean and variability of variables in the first cluster

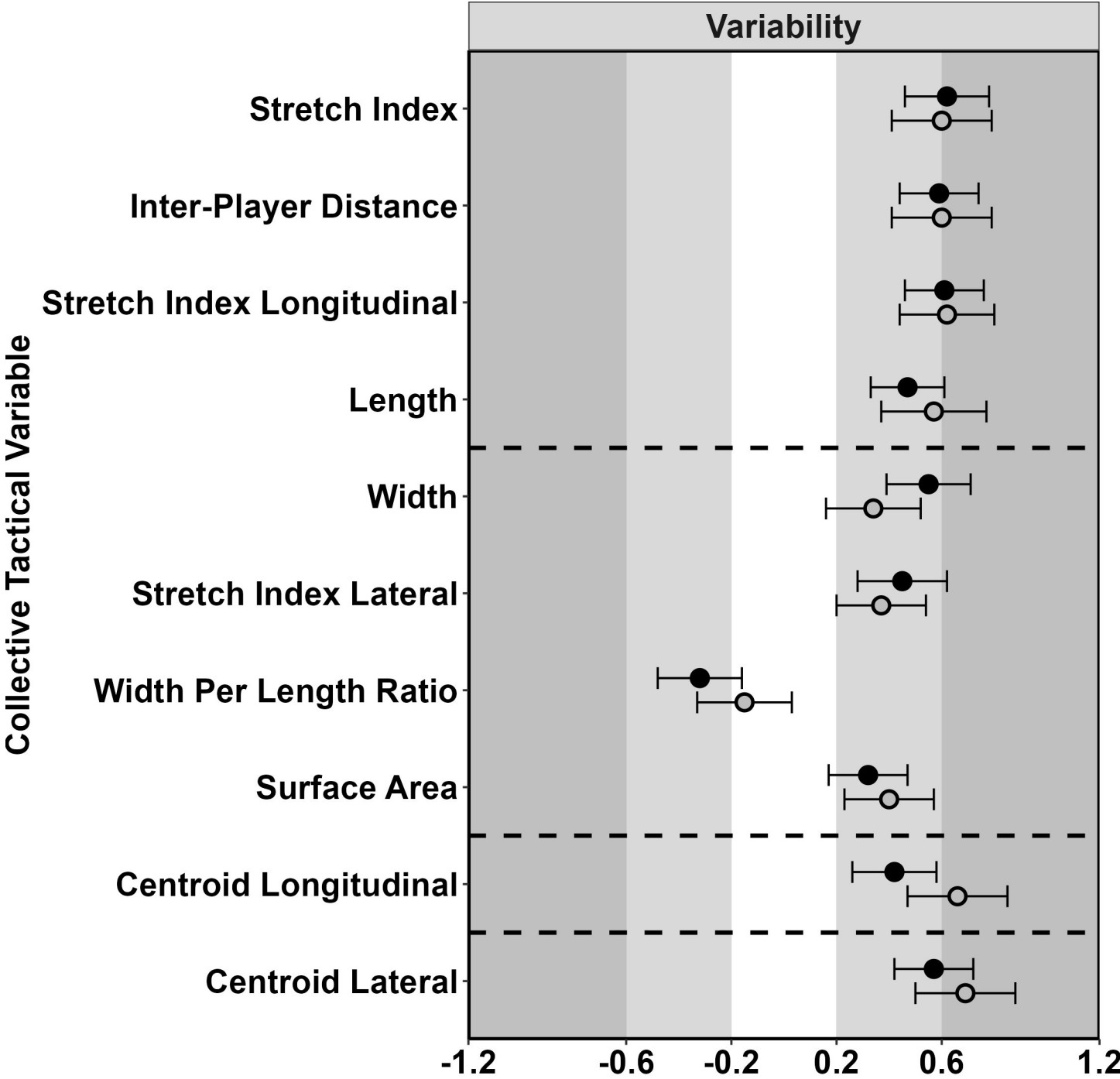

**Fig 6. Standardised effect of two SDs of possession duration for the variability on collective tactical variables.** For the team on attack (filled circles) and defence (open circles). Circles towards the left indicate higher in short possession and the right higher in long possession.

displayed small to moderate effects during defence, with the forward sub-group displaying negative sign and midcourt and defender sub-groups displaying positive.

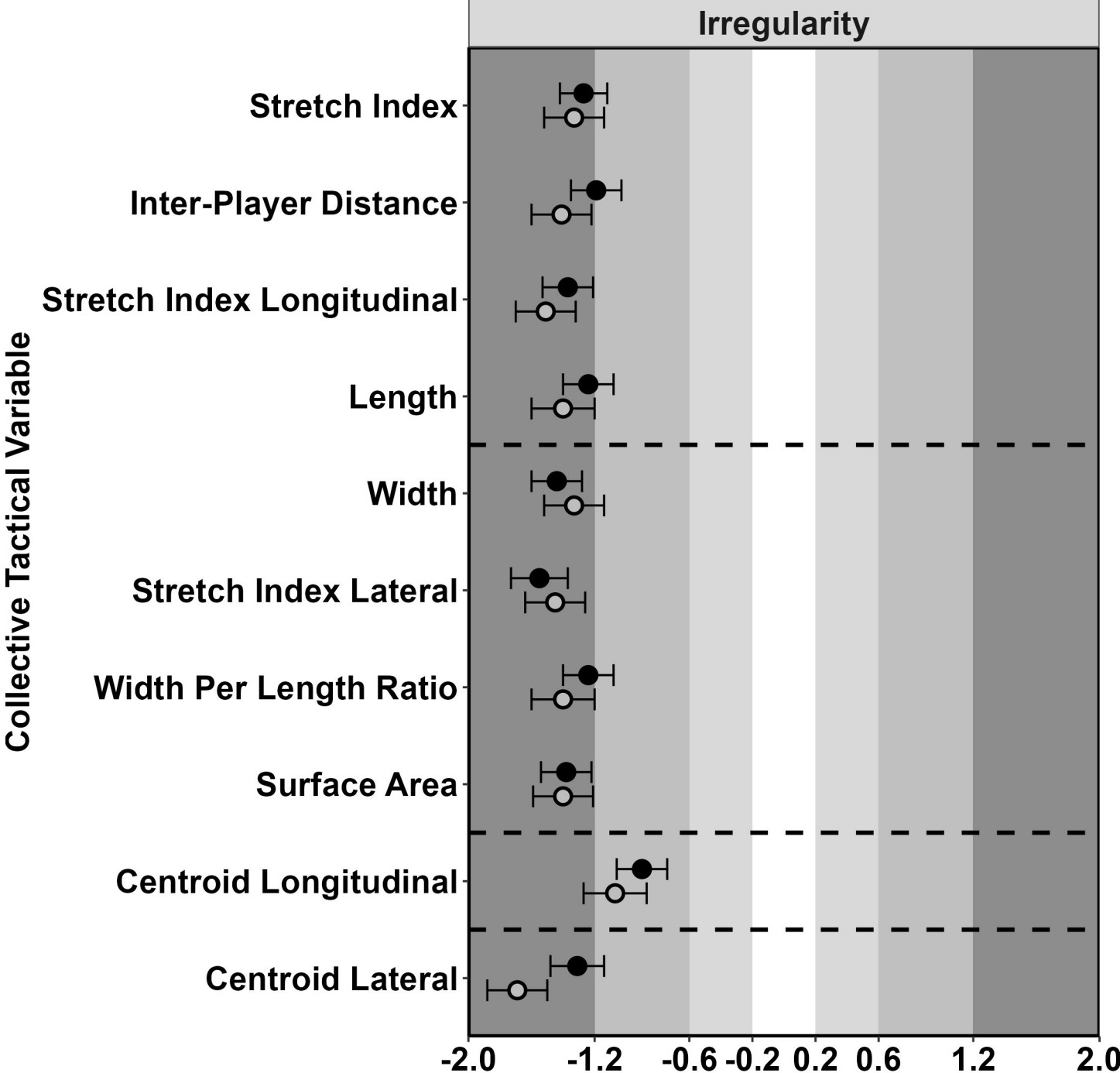

**Fig 7. Standardised effect of two SDs of possession duration for the irregularity on collective tactical variables.** For the team on attack (filled circles) and defence (open circles). Circles towards the left indicate higher in short possession and the right higher in long possession.

### Effect of match time

Substantial changes over the match in attack for the mean of variables in the first cluster occurred for the team and the forward and midcourt sub-groups [S23–S26 Tables]. During defence, similar small to moderate decreases were shown by the team and midcourt sub-

group, but not the forward subgroup, as the defender sub-group displayed similar effects instead. The variability of first cluster variables also displayed small to moderate effects for the midcourt and defender sub-groups (implying reduced variability of longitudinal dispersion over the match). Lastly, the midcourt sub-group displayed small effects for the second cluster of variables in defence.

### Effect of ladder-points difference

Mainly unclear trivial to small observed values [S27–S30 Tables], only the variability in the second cluster variables had good evidence for a small reduction in lateral dispersion of the players when matched against a weaker team. Positional sub-groups displayed similar trivial to small effects to that of the team, although there was reasonable evidence for the defenders' small substantial reduction in longitudinal regularity on attack when playing a weaker team

### Effect of season time

Similar to the effects of ladder point's difference, both clusters of collective tactical variables displayed unclear trivial to small observed values [S31–S34 Tables] over the season.

### Effects of match identity

The standard deviation representing differences between match means (after adjustment for all other effects in the model) had inadequate precision for every measure and every collective tactical variable. The observed magnitudes for the standard deviations were generally small for the team but ranged from trivial to large for the sub-groups.

### Weakly informative priors

The effect which showed the greatest shrinkage was the difference of the teams centroid longitudinal between attack and defence possessions, with a minimally informative prior the effect was 4.29, ±0.46 (mean, ±90% compatibility limits), after shrinkage with a weakly informative prior the effect became 4.24, ±0.46, which is obviously a negligible shrinkage.

## Discussion

The results show meaningful effects on collective tactical variable in netball for the team and sub-groups and several contextual factors accounted for. The main discussion points will focus on the increased lateral dispersion on attack, longitudinal dispersion on defence and greater irregularity of collective tactical variables for active sub-groups.

The correlation analyses showed that the collective tactical variables could be grouped into clusters, with higher correlations between variables within the clusters than between the clusters, and usually with the same clustering for the team, the sub-groups and the three measures (mean, SD and ApEn). The clusters represent four largely independent dimensions: longitudinal and lateral dispersion and positioning. For the sub-groups, the variables surface area and width per length ratio represent intersections between longitudinal and lateral dispersion clusters, containing elements of both.

The reason for the high correlations between some variables is obvious from the way they are calculated, as noted by Bartlett, Button [29] for inter-player distance and stretch index, which had the highest correlation in the present study. The correlations between some variables reduced in the player positional subgroups, potentially because of the effect on dispersion when the goal keeper or goal shooter are included or excluded in the subgroups. The effect of these two players on the calculations of the variables also explains why surface area and width

per length ratio displayed lower correlations with variables within the lateral dispersion cluster for sub-groups than for the team. At the team level, longitudinal dispersion is constrained by the court restrictions imposed upon the goal keeper and goal shooter positions [see Fig 1]; thus, high correlations were found for the variables surface area and width per length ratio with other variables in the lateral dispersion cluster. Additionally, the tactics of the shooters, such as a holding style of play verses a rotating circle can contribute to the dispersion of the team. At the sub-group level, one or both of the goal keeper and goal shooter are removed from the calculations, resulting in a greater contribution of longitudinal dispersion to these two variables.

The effects on variables within each cluster were generally similar, which is an obvious consequence of the high correlations between the variables within each cluster. For the purposes of this discussion, the focus will therefore be on one variable with the highest correlations in each cluster. Inter-player distance rather than stretch index was chosen as the representative measure of longitudinal dispersion, because it is more practical for coaches and players. Width was chosen as the representative measure for lateral dispersion, along with the two centroids for longitudinal and lateral positioning. These variables are the least complex and provide an easy to understand way of providing information relating to collective tactical variables.

Of all the factors modifying the collective tactical variables, possession type (representing the difference between attacking and defending possessions) had generally the largest observed effects for the team and all subgroups. The largest effects of possession type were on centroid longitudinal, a result of moving closer to the attacking goal on attack and defending goal on defence. The mostly moderate differences between attacking and defending possessions for the width of the team and of midcourt and defender sub-groups represent increased lateral dispersion during attacking possessions, which aligns with previous findings in soccer and common coaching principles surrounding the strategies of attack and defence in team sports [24, 26, 27, 48, 80]. The attacking team players may focus on destabilising the defending opposition through increased lateral dispersion [24, 26]. The midcourt sub-group most epitomizes the use of lateral dispersion on attack, an affordance of court structure and positional restrictions, whereby midcourt players are not permitted in the shooting circle. For coaches this is may be useful as a benchmark of how certain strategies could be performed.

Positional sub-groups displayed several differing collective tactical variables results influenced by possession type. Forward and defender sub-groups showed moderate to large greater inter-player distance irregularity during their active phases of a match (attacking possessions for forwards and defensive possessions for defenders). Defenders also showed moderately greater width irregularity during defence, possibly because the activities of off-ball guarding and defending result in more irregularity as the defending sub-group players jostle for position and react to their direct opponents' movements [69]. Additionally, Australian netball teams have a preference to play one-on-one defence over zone defence, further emphasizing the dyadic and reactive nature of netball defending [81]. During non-active phases, both the forward and defender sub-groups exhibited less irregularity of collective tactical variables and large to very large mean differences in inter-player distances as they spread out to provide mostly stationary assistance to their active teammates. This is influenced by court restrictions imposed upon players, as well as the distinctive movement features most common in netball, whereby walking with straight movement and neutral acceleration are most prevalent [45].

Several small effects were found for the differences between possession outcomes (scoring and turnover). When the home team and opposition team possessions resulted in a score, the home team displayed increased mean values and increased variability of longitudinal dispersion, suggesting that expansion and variability through the length of the court is also an important attacking principle to stretch defenders apart. This principle is further supported by the

forward sub-group displaying small observed decreases in inter-player distances and the mid-court sub-group reduced inter-player distance irregularity when the home team scored. Thus, successful possessions may tend to be those with a more direct path to the goal, where opposition defenses are destabilised and separated [53]. When the opposition scored, the defender sub-group presented small reductions in inter-player distances, width and surface area, representing compactness of positioning, previously highlighted as a critical defensive strategy in netball [53]. Of note, while teams still score with defenders being compact, these strategies may be more effective against slower teams and not during turnover possessions.

Evolution of some collective tactical variables over the course of a match was apparent in this study. A reduction in the longitudinal dispersion of the team, midcourt and forward sub-groups during attack and of the team, midcourt and defender sub-groups during defence aligns with previous findings in soccer, where team dispersion and field positioning decreased in the second half [60]. The effect on the midcourt sub-group showed that these players adopt closer positioning to one another as the match progresses. Fatigue may at least partly explain these changes during a match, as previous research found reductions in intensity (significant decreases of heart rate max >85%) between the first and second halves [82]. Which could also contribute to less movement and shorter passes of the ball.

As matches progressed, the forward sub-group displayed increased mean surface area and increased variability of inter-player distance during defence, highlighting the warning of [83] that analysis of only team-level behaviours can fail to capture important sub-group behaviours. Especially if different team tactics are employed both for the shooters and their defenders. Limitations of team-level analysis only is further exacerbated in netball owing to the positional restrictions, causing collective tactical variables to be sometimes anchored with the GS or GK (positions with most restricted movement). Similar findings have been found in soccer where the goal keeper is predominantly stationary compared to out-field players and thus frequently omitted from calculation of collective tactical variables [84]. The small to moderate increases of variability with increasing possession time for most collective tactical variables for the team and all sub-groups simply reflect more opportunity for players to disperse longitudinally and laterally as the possession progresses [25]. The moderate to large reductions in irregularity of all collective tactical variables for the team and all sub-groups are also an expected outcome, as longer possessions allow the time series to cancel out noise and chaotic patterns that are prevalent in shorter sequences [36, 58, 74, 85].

The strength of opposition has previously been shown in soccer to influence collective tactical variables, with greater surface area against stronger opponents and shorted possession lengths against weaker opponents [23, 58]. In the current study of only seven matches, opposition strength (ladder-points difference) had a limited number of clear effects, but they were generally consistent with the findings of [23]. The clearest effect of ladder-points difference occurred in the defender sub-group on attack, where the small reductions in irregularity of longitudinal dispersion (in particular, stretch index) when playing a weaker team is consistent with the weaker team's inability to disrupt opposition defenders.

Changes in collective tactical variables over the season also had limited clear effects, owing to the small number of matches: only the team showed clear small increases of width. Season trends are still of potential importance for future research, as previous literature in soccer has suggested that tactics and group behaviours over the course of a season change, as players adjust to the team strategy and further coordinate their behaviours with teammates [11].

Score state had the most evidence for trivial effects for the team but there were clear substantial effects for some of the measures in some of the sub-groups. The increased longitudinal dispersion displayed by the mid-court sub-group on attack and defence suggests affordances of increased space lengthwise down the court may be apparent when leading on the

scoreboard. The forward sub-group's reduction in longitudinal dispersion when leading during defence is potentially an adaptive behaviour in response to the midcourt sub-group utilizing more space closer to the attacking goal. The trivial effects for variability of lateral dispersion for the midcourt and defender sub-groups when leading suggest these sub-groups retain the same defensive positioning that had successfully placed them in a winning situation. Previous research into score state in soccer games is hard to compare to netball, as in soccer the difficulty to score goals means soccer teams are better able to defend a winning possession [86]. While in netball the equal share of possession after a goal is scored allows for more opportunities for both teams to score.

Finally, the random effect of match ID had inadequate precision for all collective tactical variables, a consequence of the small number of matches. The observed substantial effects are consistent with factors effecting match means that were not accounted for in the model.

## Limitations / future research directions

High correlations of collective tactical variables in elite netball have been identified in this research, presenting several variables that can be a focus for future study. However, several other considerations must be accounted for when using these variables. Collective tactical variables are a reduction analysis technique, whereby large amounts of information are reduced to one dimension [12]. As such, future research should consider other dimensions such as the behaviours of the opposing team, inter-team coordination and line forces; which provides a measure of inter-team sub-group cohesion [28]. To better assess the effects of ladder-points difference and season time, many more matches than seven would be required. More matches, from multiple teams and locations would also improve the precision of the effect of match identity and thereby reveal whether there are factors affecting the overall tactics in each match other than those assessed in the present study (e.g. coaches changing strategy for specific opponents). The findings of the present study could be implemented in the training environment and in matches.

## Conclusion

Team and sub-group results have identified how collective tactical variables dynamically shift over the course of a match and during different phases. Additionally, the contribution of correlation analysis further enhances the understanding of how collective tactical variables can be implemented into elite netball and highlights those variables which may be most suited or redundant for future analysis. Finally, exploration at the team and sub-group level, has displayed interactions among each of their components and how specific positions can influence organisation levels of higher order. Through the exploration of collective tactical variables, an initial base of knowledge has been provided to support the implementation of this type of analysis in elite netball.

## Supporting information

**S1 Video. Video animations of the collective tactical variables for one possession.** For each sub-group.
(MP4)

**S2 Video. Video animations of the collective tactical variables for one possession.** For each sub-group.
(GIF)

**S1 Table. Correlations between collective tactical variables for the forward sub-group on attack.** Data shown are the range of the three correlations (mean, standard deviation and entropy). The variables have been ordered and outlined to show clusters with generally higher correlations between variables within the clusters than between the clusters.
(PDF)

**S2 Table. Correlations between collective tactical variables for the forward sub-group on defence.** Data shown are the range of the three correlations (mean, standard deviation and entropy). The variables have been ordered and outlined to show clusters with generally higher correlations between variables within the clusters than between the clusters.
(PDF)

**S3 Table. Correlations between collective tactical variables for the midcourt sub-group on attack.** Data shown are the range of the three correlations (mean, standard deviation and entropy). The variables have been ordered and outlined to show clusters with generally higher correlations between variables within the clusters than between the clusters.
(PDF)

**S4 Table. Correlations between collective tactical variables for the midcourt sub-group on defence.** Data shown are the range of the three correlations (mean, standard deviation and entropy). The variables have been ordered and outlined to show clusters with generally higher correlations between variables within the clusters than between the clusters.
(PDF)

**S5 Table. Correlations between collective tactical variables for the defender sub-group on attack.** Data shown are the range of the three correlations (mean, standard deviation and entropy). The variables have been ordered and outlined to show clusters with generally higher correlations between variables within the clusters than between the clusters.
(PDF)

**S6 Table. Correlations between collective tactical variables for the defender sub-group on defence.** Data shown are the range of the three correlations (mean, standard deviation and entropy). The variables have been ordered and outlined to show clusters with generally higher correlations between variables within the clusters than between the clusters.
(PDF)

**S7 Table. Simple statistics provided by the mixed model (predicted mean for the middle of a match and mean possession duration; residual within-match standard deviation) for the team and positional sub-groups on attack.** With the exception of the mean centroid longitudinal and lateral, the statistics were derived via log-transformation, hence SDs are shown as times/divide factors. Clusters of variables representing consecutively longitudinal dispersion, lateral dispersion, longitudinal position and lateral position are outlined.
(PDF)

**S8 Table. Simple statistics provided by the mixed model (predicted mean for the middle of a match and mean possession duration; residual within-match standard deviation) for the team and positional sub-groups on defence.** With the exception of the mean centroid longitudinal and lateral, the statistics were derived via log-transformation, hence SDs are shown as times/divide factors. Clusters of variables representing consecutively longitudinal dispersion, lateral dispersion, longitudinal position and lateral position are outlined.
(PDF)

**S9 Table. Collective tactical variables for the forward sub-group attack and defence possessions adjusted for possession duration, score difference, ladder point's difference, match trend and season trend.** Dashed lines separate the variables into the clusters defined by the correlations in Table 2. With the exception of the mean centroid longitudinal and lateral, the statistics were derived via log-transformation, hence SDs are shown as percents. Data for attack and defence are predicted means (raw units) from the mixed model, and SDs are an appropriate residual representing differences between possessions. Data for attack minus defence are predicted mean differences (% units) with 90% compatibility limits (% units) and decisions about the magnitude of the differences.
(PDF)

**S10 Table. Collective tactical variables for the midcourt sub-group attack and defence possessions adjusted for possession duration, score difference, ladder point's difference, match trend and season trend.** Dashed lines separate the variables into the clusters defined by the correlations in Table 2. With the exception of the mean centroid longitudinal and lateral, the statistics were derived via log-transformation, hence SDs are shown as percents. Data for attack and defence are predicted means (raw units) from the mixed model, and SDs are an appropriate residual representing differences between possessions. Data for attack minus defence are predicted mean differences (% units) with 90% compatibility limits (% units) and decisions about the magnitude of the differences.
(PDF)

**S11 Table. Collective tactical variables for the defender sub-group attack and defence possessions adjusted for possession duration, score difference, ladder point's difference, match trend and season trend.** Dashed lines separate the variables into the clusters defined by the correlations in Table 2. With the exception of the mean centroid longitudinal and lateral, the statistics were derived via log-transformation, hence SDs are shown as percents. Data for attack and defence are predicted means (raw units) from the mixed model, and SDs are an appropriate residual representing differences between possessions. Data for attack minus defence are predicted mean differences (% units) with 90% compatibility limits (% units) and decisions about the magnitude of the differences.
(PDF)

**S12 Table. Difference between home score and home turnover (on attack) and between opposition score and opposition turnover (on defence) for each of the derived measures of each collective tactical variable for the forward sub-group.** With the exception of the mean centroid longitudinal and lateral, the statistics were derived via log-transformation, hence data are the predicted changes (%, ±90% compatibility limits) and decisions about the magnitude of the changes.
(PDF)

**S13 Table. Difference between home score and home turnover (on attack) and between opposition score and opposition turnover (on defence) for each of the derived measures of each collective tactical variable for the midcourt sub-group.** With the exception of the mean centroid longitudinal and lateral, the statistics were derived via log-transformation, hence data are the predicted changes (%, ±90% compatibility limits) and decisions about the magnitude of the changes.
(PDF)

**S14 Table. Difference between home score and home turnover (on attack) and between opposition score and opposition turnover (on defence) for each of the derived measures of**

**each collective tactical variable for the defender sub-group.** With the exception of the mean centroid longitudinal and lateral, the statistics were derived via log-transformation, hence data are the predicted changes (%, ±90% compatibility limits) and decisions about the magnitude of the changes.
(PDF)

**S15 Table. Effect of two SD of possession length (factor increases of 2.4 on attack and 2.7 on defence) on collective tactical variables for the team on attack and defence.** With the exception of the mean centroid longitudinal and lateral, the statistics were derived via log-transformation, hence data are the predicted changes (%, ±90% compatibility limits) and decisions about the magnitude of the changes.
(PDF)

**S16 Table. Effect of two SD of possession length (factor increases of 2.4 on attack and 2.7 on defence) on collective tactical variables for the forward's sub-group on attack and defence.** With the exception of the mean centroid longitudinal and lateral, the statistics were derived via log-transformation, hence data are the predicted changes (%, ±90% compatibility limits) and decisions about the magnitude of the changes.
(PDF)

**S17 Table. Effect of two SD of possession length (factor increases of 2.4 on attack and 2.7 on defence) on collective tactical variables for the midcourt's sub-group on attack and defence.** With the exception of the mean centroid longitudinal and lateral, the statistics were derived via log-transformation, hence data are the predicted changes (%, ±90% compatibility limits) and decisions about the magnitude of the changes.
(PDF)

**S18 Table. Effect of two SD of possession length (factor increases of 2.4 on attack and 2.7 on defence) on collective tactical variables for the defender's sub-group on attack and defence.** With the exception of the mean centroid longitudinal and lateral, the statistics were derived via log-transformation, hence data are the predicted changes (%, ±90% compatibility limits) and decisions about the magnitude of the changes.
(PDF)

**S19 Table. Effect of a +10 points score difference on collective tactical variables for the team on attack and defence.** With the exception of the mean centroid longitudinal and lateral, the statistics were derived via log-transformation, hence data are the predicted changes (%, ±90% compatibility limits) and decisions about the magnitude of the changes.
(PDF)

**S20 Table. Effect of a +10 points score difference on collective tactical variables for the forward's sub-group on attack and defence.** With the exception of the mean centroid longitudinal and lateral, the statistics were derived via log-transformation, hence data are the predicted changes (%, ±90% compatibility limits) and decisions about the magnitude of the changes.
(PDF)

**S21 Table. Effect of a +10 points score difference on collective tactical variables for the midcourt's sub-group on attack and defence.** With the exception of the mean centroid longitudinal and lateral, the statistics were derived via log-transformation, hence data are the predicted changes (%, ±90% compatibility limits) and decisions about the magnitude of the changes.
(PDF)

**S22 Table. Effect of a +10 points score difference on collective tactical variables for the defender's sub-group on attack and defence.** With the exception of the mean centroid longitudinal and lateral, the statistics were derived via log-transformation, hence data are the predicted changes (%, ±90% compatibility limits) and decisions about the magnitude of the changes.
(PDF)

**S23 Table. Change in collective tactical variables over a match for the team on attack and defence.** With the exception of the mean centroid longitudinal and lateral, the statistics were derived via log-transformation, hence data are the predicted changes (%, ±90% compatibility limits) and decisions about the magnitude of the changes.
(PDF)

**S24 Table. Change in collective tactical variables over a match for the forward's sub-group on attack and defence.** With the exception of the mean centroid longitudinal and lateral, the statistics were derived via log-transformation, hence data are the predicted changes (%, ±90% compatibility limits) and decisions about the magnitude of the changes.
(PDF)

**S25 Table. Change in collective tactical variables over a match for the midcourt's sub-group on attack and defence.** With the exception of the mean centroid longitudinal and lateral, the statistics were derived via log-transformation, hence data are the predicted changes (%, ±90% compatibility limits) and decisions about the magnitude of the changes.
(PDF)

**S26 Table. Change in collective tactical variables over a match for the defender's sub-group on attack and defence.** With the exception of the mean centroid longitudinal and lateral, the statistics were derived via log-transformation, hence data are the predicted changes (%, ±90% compatibility limits) and decisions about the magnitude of the changes.
(PDF)

**S27 Table. Effect of the strongest opposition minus the weakest opposition on collective tactical variables for the team on attack and defence.** With the exception of the mean centroid longitudinal and lateral, the statistics were derived via log-transformation, hence data are the predicted changes (%, ±90% compatibility limits) and decisions about the magnitude of the changes.
(PDF)

**S28 Table. Effect of the strongest opposition minus the weakest opposition on collective tactical variables for the forward's sub-group on attack and defence.** With the exception of the mean centroid longitudinal and lateral, the statistics were derived via log-transformation, hence data are the predicted changes (%, ±90% compatibility limits) and decisions about the magnitude of the changes.
(PDF)

**S29 Table. Effect of the strongest opposition minus the weakest opposition on collective tactical variables for the midcourt's sub-group on attack and defence.** With the exception of the mean centroid longitudinal and lateral, the statistics were derived via log-transformation, hence data are the predicted changes (%, ±90% compatibility limits) and decisions about the magnitude of the changes.
(PDF)

**S30 Table. Effect of the strongest opposition minus the weakest opposition on collective tactical variables for the defender's sub-group on attack and defence.** With the exception of the mean centroid longitudinal and lateral, the statistics were derived via log-transformation, hence data are the predicted changes (%, ±90% compatibility limits) and decisions about the magnitude of the changes.
(PDF)

**S31 Table. Change in collective tactical variables over the season for the team on attack and defence.** With the exception of the mean centroid longitudinal and lateral, the statistics were derived via log-transformation, hence data are the predicted changes (%, ±90% compatibility limits) and decisions about the magnitude of the changes.
(PDF)

**S32 Table. Change in collective tactical variables over the season for the forward's sub-group on attack and defence.** With the exception of the mean centroid longitudinal and lateral, the statistics were derived via log-transformation, hence data are the predicted changes (%, ±90% compatibility limits) and decisions about the magnitude of the changes.
(PDF)

**S33 Table. Change in collective tactical variables over the season for the midcourt's sub-group on attack and defence.** With the exception of the mean centroid longitudinal and lateral, the statistics were derived via log-transformation, hence data are the predicted changes (%, ±90% compatibility limits) and decisions about the magnitude of the changes.
(PDF)

**S34 Table. Change in collective tactical variables over the season for the defender's sub-group on attack and defence.** With the exception of the mean centroid longitudinal and lateral, the statistics were derived via log-transformation, hence data are the predicted changes (%, ±90% compatibility limits) and decisions about the magnitude of the changes.
(PDF)

## Acknowledgments

The authors would like to thanks the athletes and staff who contributed to the study.

## Author Contributions

**Conceptualization:** Ryan W. Hodder, Kevin A. Ball, Jamie Bahnisch, Fabio R. Serpiello.

**Data curation:** Ryan W. Hodder, Kevin A. Ball, Fabio R. Serpiello.

**Formal analysis:** Ryan W. Hodder, Will G. Hopkins.

**Funding acquisition:** Fabio R. Serpiello.

**Investigation:** Ryan W. Hodder, Will G. Hopkins, Kevin A. Ball, Fabio R. Serpiello.

**Methodology:** Ryan W. Hodder, Will G. Hopkins, Kevin A. Ball, Jamie Bahnisch, Fabio R. Serpiello.

**Project administration:** Ryan W. Hodder, Fabio R. Serpiello.

**Resources:** Ryan W. Hodder, Kevin A. Ball, Fabio R. Serpiello.

**Software:** Ryan W. Hodder, Will G. Hopkins, Kevin A. Ball, Fabio R. Serpiello.

**Supervision:** Ryan W. Hodder, Will G. Hopkins, Kevin A. Ball, Jamie Bahnisch, Fabio R. Serpiello.

**Validation:** Will G. Hopkins, Kevin A. Ball, Fabio R. Serpiello.

**Visualization:** Ryan W. Hodder, Fabio R. Serpiello.

**Writing – original draft:** Ryan W. Hodder.

**Writing – review & editing:** Ryan W. Hodder, Will G. Hopkins, Kevin A. Ball, Fabio R. Serpiello.

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
