## [Decision Letter · Decision Letter 0]

28 Feb 2023

PONE-D-23-00953Exploration of collective tactical variables in elite netball: an analysis of team and sub-group positioning behavioursPLOS ONE

Dear Dr. Hodder,

Thank you for submitting your manuscript to PLOS ONE. After careful consideration, we feel that it has merit but does not fully meet PLOS ONE’s publication criteria as it currently stands. Therefore, we invite you to submit a revised version of the manuscript that addresses the points raised during the review process.

We look forward to receiving your revised manuscript.

Kind regards,

Bruno Gonçalves

Academic Editor

PLOS ONE

Journal Requirements:

"RH is supported by a scholarship partially funded by Netball Victoria."

"RH is supported by a scholarship partially funded by Netball Victoria.

https://vic.netball.com.au/

No role in study design, data collection and analysis, decision to publish, or preparation"

6. Please ensure that you include a title page within your main document. You should list all authors and all affiliations as per our author instructions and clearly indicate the corresponding author.

Reviewers' comments:

Reviewer's Responses to Questions

**Comments to the Author**

1. Is the manuscript technically sound, and do the data support the conclusions?

Reviewer #1: Yes

2. Has the statistical analysis been performed appropriately and rigorously? 

Reviewer #1: Yes

3. Have the authors made all data underlying the findings in their manuscript fully available?

Reviewer #1: Yes

4. Is the manuscript presented in an intelligible fashion and written in standard English?

Reviewer #1: Yes

5. Review Comments to the Author

Reviewer #1: Thank you for providing the opportunity to review this interesting study. The authors provide a novel analysis of collective tactical variables in netball with advanced analysis techniques providing insight into the positioning behaviors of the team and positional groups. Whilst some rationale is provided for this, linking to research in other sports, my major comment about the manuscript is the lack of practical application, or future research directions as an outcome of the findings. Additionally, I believe the presentation of the results/discussion could be improved to improve clarity to the reader – I understand it is hard to present and discuss so much data, but by stating in the discussion that only going to discuss some variables suggests the others aren’t important - therefore I question why they are included in the manuscript. Please see below for specific comments:

Abstract

L3 – please also state positional spatial restrictions alongside unable to move with the ball. If not space for this, I would remove the bracket as it suggests that is the only restriction.

L6 – please change national-level to elite. You use elite in the title. Or use professional throughout

Introduction

L31 – please add a full stop at the end of the sentence.

L44 – please remove the full stop before the citation.

L46-47 – should this be the same paragraph? Please address paragraph structure as lines 47-49 shouldn’t be a stand a-lone paragraph.

L57 – ‘and in Australian rules..’ should this be ‘in Australian rules..’

L63 – please clarify here what you mean by ‘marker of performance’. What performance outcome?

L68 – please add a full stop at the end of the sentence.

L68-69 – again, should this be a separate paragraph?

L74 – please can you expand on why important

L79 – you start to use ApEn here. I assume this is an abbreviation for approximate entropy but you haven’t identified it and then later go back to using the full write out. Please can you clarify in text.

L81 – please add a full stop at the end of the sentence.

L89 – please add ‘positional’ to the court restrictions statement i.e., positional court restrictions

Lines 96 – 101 – Please make clear in this section that the studies cited aren't in netball - given the paragraph starts on research in netball it reads as if these are in netball too.

Line 102 – could you add a sentence stating these are yet to be investigated in netball. This would lead into the aims better.

Line 104 – it becomes more clear later in the paper but at this stage it isn’t clear what you mean by sub-groups please make this more clear eg. By stating the groups in brackets.

Methods

L111 please amend sentence. You state participants twice. Please also add in that they are females.

L110-11 as per comment in abstract – clarity and consistency in terminology of language use for playing level. At the moment you have used three different descriptors – national, elite and professional.

L120 please can you provide information of positional sub-groups here and sample size for each group.

Event data section

Please refer to and use the recent consensus for definitions - particularly for clarity on use of ‘Turnover’

https://bjsm.bmj.com/content/early/2023/02/07/bjsports-2022-106187.abstract?casa_token=Tg_fKl9PXFQAAAAA:YIFltSYgiE2-X2AHTLAUc0iyF9UkoKQjaw3L-hYm1uHeX6ygkn17IQTdAbmbJ00aqkZUDIov7iU

L149 – can you give a couple examples of ‘other’

Data processing section

Can you provide the R packages used for different stages of analysis.

L170 – please remove the full stop before the citation.

L171 – more detail could be provided here on what the points mean – however I am aware it is already a long paper!

L174-176 – positional grouping is unclear. Why have you included WA and WD in multiple groups. You have cited a previous study but they haven’t done this they have actually analysed as separate positions but discussed the, in defensive and attacking groups with Centre as stand alone. Please provide further justification for this. Please also consider the term ‘forwards’ this isn’t really used in current netball literature, could you change to attackers given you have defenders?

L238 – ‘were’ used in the analysis

L304 – please ensure consistency in capital or lower case P

Results

You have done a great job at a difficult task of summarising and presenting such a large amount of data. However, I do wonder whether some of what is in supplementary is included in text – particularly the sub-group analysis given it is discussed a lot. Is there any way you could combine some tables/ figures? I appreciate this is easier said than done!

Table 3 and 4. Can you enter a new line post the table title for the information with it to make the table title more clear.

Is the statement on ‘with the exception of the mean centroid….’ Needed given you left statistics in the methods.

L363 as stated above given the aim was to investigate sub-groups and team I would like to see more results for the sub-groups in-text not supplementary.

L393 if tables have to remain supplementary could you provide some data in text to support.

Figures 5-7

You don’t refer to figures 5, 6 or 7 in text in the possession section? This would likely overcome my comment about data in text if there are figures to support as well.

Please also consider merging these into one figure and having as figure 5a,b,c

Discussion

L449 Can you recap the aim of the study here please first

L450-452 - Why? If the rest isn’t going to be fully discussed, why is it presented and included?

L463 Please consider the use of the word apparently here - if it is evident in the data state that, if it is speculative state that

L468 More explanation here - figure 1 doesn’t show this clearly. Readers who don’t know netball but are reading from a methodological perspective would need to understand the restrictions of these positions.

Also consider here that this is one team only over 7 matches and that they may have been playing a holding GS for example rather than rotating circle (as highlighted earlier as a consideration) - therefore acknowledging a limitation of playing style is important

L472 Please summarise the findings with practical or future research application. It currently isn’t clear to what this means and how it can be used

L473 Be consistent with terminology below you state positional tactical variables - please do the same here so it is clear discussing the same thing given the large amount of data/variables in the study

L488 Again, this use of apparently isn’t clear

L491 So what can coaches do with this? You state most practical for coaches and players in the above paragraph so I would like to see more practical applications provided.

L524-526 Expand further - therefore less movement off the ball and shorter passes?

L520 – brackets for citation date

L532 please add consideration of different playing styles vs different oppositions based on shooting circle

L533 – you state similar findings have been found but don’t provide a citation – please add.

L582 – again, how? This would be okay if provided examples in discussion above.

6. PLOS authors have the option to publish the peer review history of their article (what does this mean?). If published, this will include your full peer review and any attached files.

Reviewer #1: **Yes: **Dr Sarah Whitehead

---

## [Author Response · Author response to Decision Letter 0]

13 Apr 2023

Please see attached Response to Reviewers document for my responses to reviewers comments.

---

## [Decision Letter · Decision Letter 1]

5 Jun 2023

PONE-D-23-00953R1Exploration of collective tactical variables in elite netball: an analysis of team and sub-group positioning behavioursPLOS ONE

Dear Dr. Hodder,

Thank you for submitting your manuscript to PLOS ONE. After careful consideration, we feel that it has merit but does not fully meet PLOS ONE’s publication criteria as it currently stands. Therefore, we invite you to submit a revised version of the manuscript that addresses the points raised during the review process.

We look forward to receiving your revised manuscript.

Kind regards,

Bruno Gonçalves

Academic Editor

PLOS ONE

Journal Requirements:

Reviewers' comments:

Reviewer's Responses to Questions

**Comments to the Author**

1. If the authors have adequately addressed your comments raised in a previous round of review and you feel that this manuscript is now acceptable for publication, you may indicate that here to bypass the “Comments to the Author” section, enter your conflict of interest statement in the “Confidential to Editor” section, and submit your "Accept" recommendation.

Reviewer #1: All comments have been addressed

2. Is the manuscript technically sound, and do the data support the conclusions?

Reviewer #1: Yes

3. Has the statistical analysis been performed appropriately and rigorously? 

Reviewer #1: Yes

4. Have the authors made all data underlying the findings in their manuscript fully available?

Reviewer #1: Yes

5. Is the manuscript presented in an intelligible fashion and written in standard English?

Reviewer #1: Yes

6. Review Comments to the Author

Reviewer #1: Thank you to the authors for the revisions. All comments have been addressed to my satisfaction. There a couple of minor edits need addressing before accepting, see below:

Line 76 please address sentence structure – the sentence added doesn’t flow on its own and should be part of the prior sentence.

Line 94 consider re wording ‘inability of the player in possession of the ball to move’ – technically they can move, just not replace their landing foot – therefore at the elite level they frequently still move 1m or so in possession of the ball

L169 space needed between . and utilizing.

L186 I understand your response to my comment about the positional sub-groups, however this statement is still incorrect as it the groups you have used and how you have termed them isn’t in-line with previous research with some positions in multiple groups and attackers being termed forwards. Please either adapt this sentence to acknowledge this.

L472 ‘subgroups’ – throughout the rest of the paper it is written as sub-groups except this sentence. Please change and check the rest of the manuscript.

L543 – please check the flow of the last two sentences and consider combining them.

7. PLOS authors have the option to publish the peer review history of their article (what does this mean?). If published, this will include your full peer review and any attached files.

Reviewer #1: **Yes: **Dr Sarah Whitehead

---

## [Author Response · Author response to Decision Letter 1]

31 Oct 2023

See response to reviewers document attached in step 2

---

## [Editor Report · Decision Letter 2]

30 Nov 2023

Exploration of collective tactical variables in elite netball: an analysis of team and sub-group positioning behaviours

PONE-D-23-00953R2

Dear Dr. Hodder,

We’re pleased to inform you that your manuscript has been judged scientifically suitable for publication and will be formally accepted for publication once it meets all outstanding technical requirements.

Kind regards,

Bruno Gonçalves

Academic Editor

PLOS ONE
---

## [Editor Report · Acceptance letter]

16 Feb 2024

PONE-D-23-00953R2 

PLOS ONE

Dear Dr. Hodder, 

I'm pleased to inform you that your manuscript has been deemed suitable for publication in PLOS ONE. Congratulations! Your manuscript is now being handed over to our production team.

Kind regards, 

on behalf of

Dr. Bruno Gonçalves 

Academic Editor

PLOS ONE